# Dietary Methionine Increased the Growth Performances and Immune Function of Partridge Shank Broilers after Challenged with Coccidia

**DOI:** 10.3390/ani13040613

**Published:** 2023-02-09

**Authors:** Anqiang Lai, Zehong Yuan, Zhongcheng Wang, Binlong Chen, Li Zhi, Zhiqiu Huang, Yi Zhang

**Affiliations:** 1College of Animal Science, Xichang University, Xichang 615000, China; 2Animal Nutrition Institute, Sichuan Agricultural University, Chengdu 611130, China

**Keywords:** methionine, partridge shank broilers, requirements, coccidia, immune function

## Abstract

**Simple Summary:**

Met is the first limiting amino acid in the poultry diet. In this study, optimal levels of dietary Met improved the production of partridge shank broilers. This result might be related to immune function and antioxidant capacity. Moreover, the requirements of digestible Met in terms of on-production performance (ADG and F/G) and immune function (sIgA in ileum mucosa) in partridge shank broilers (1–21 day) were calculated to be 0.418, 0.451, and 0.451% of diet, respectively, when birds were given anticoccidial drug treatment, with corresponding figures of 0.444, 0.455, and 0.452 when coccidia vaccine was administered.

**Abstract:**

The present study investigated the effects of methionine (Met) on growth, immune function, and antioxidant capacity in partridge shank broilers, which were treated with either an anticoccidial drug or a coccidia vaccine. Chickens were fed five graded levels of Met (0.33%, 0.39%, 0.45%, 0.51%, or 0.57%) for 21 days in combination with the drug or vaccine. The results revealed that an optimal level of Met supplementation (1) increased ADFI (average daily feed intake), ADG (average daily gain), and F/G values (feed-to-gain ratio), indicating improved production; (2) increased OPG levels (oocysts per gram feces), intestinal lesion scores, bursa of Fabricius and thymus indexes, and sIgA content; (3) improved GSH-Px activities, and increased content levels of T-protein, albumin, and urea nitrogen. In addition, birds in the anticoccidial drug group had higher final weights, higher ADFI and ADG values, as well as lower F/G values, compared with birds in the vaccine group, indicating that coccidia vaccine reduces the performance of broilers. In conclusion, we found that an optimal level of dietary Met improved the production of partridge shank broilers, and this result might be related to immune function and antioxidant capacity. Optimal levels of digestible Met in terms of production performance (ADG and F/G) and immune function (sIgA in ileum mucosa) in partridge shank broilers (1–21 days) were found to be 0.418, 0.451, and 0.451 of diet, respectively, when birds were given anticoccidial drug treatment, with corresponding figures of 0.444, 0.455, and 0.452% when the coccidia vaccine was administered.

## 1. Introduction

The partridge shank broiler is a native Chinese breed of chicken which has the advantage of a low feed-to-weight-gain ratio and a good meat quality that is popular among consumers [1]. However, broilers are host to several deadly diseases that hamper productivity and harm welfare, leading to high levels of infection and, in some cases, high mortality rates [2]. These include Eimeria infections such as coccidiosis, which is an intestinal disease caused by protozoan parasites of the genus. In chickens, coccidiosis can damage the intestinal structure and reduce digestion and absorption functions, leading to reduced production performance and even death [3,4]. At present, the main methods to control coccidia involve anticoccidial drugs or a coccidia vaccine. Inhibition of coccidia is achieved by killing coccidiasis oocysts. The immune response of the coccidia vaccine in-volves both cellular immunity and humoral immunity [5,6]. Under these two control schemes, the bodies of birds are subject to different physiological conditions that may affect nutrient usage. Grimble and Grimble (1998) found that the immunoregulatory actions of vaccines exert a considerable impact on methionine (Met) metabolism [7]. Met acts as the first limiting amino acid in maize–soybean diets. In poultry, Met plays a major role in the immune system response and in the antioxidant defense system, leading to improved health, growth, and development in birds [8]. However, there have been no studies to date on the Met requirements of chickens under the two coccidia control schemes. It is necessary, therefore, to evaluate the Met requirements of chickens in different immune states.

In poultry, mucosal immunity is the first immune barrier against the invasion of exogenous microorganisms. CD4 cells, CD8 cells, and IgA are all involved in this process. CD4 cells mainly play a supporting role, helping CD8 cells to produce an effective immune response against viruses and bacteria, killing pathogenic microorganisms [9,10]. IgA plays a very important role in microbial invasion [11]. Changes in CD4, CD8, and IgA levels can therefore reflect changes in the immune activities of chickens. In addition, re-searchers have found that, in chickens infected with coccidiosis, ROSs (reactive oxygen species) were produced in resistance to microorganisms attacking the intestinal epithelium [12,13]. However, excessive oxidative stress can also cause intestinal damage.

To date, there have been few reports on the effects of Met on the antioxidant capacity and immune function of chickens after infection with Eimeria, and there is a notable lack of research concerning partridge shank broilers. However, it is known that Met plays a significant role as a substrate in protein synthesis, including the synthesis of other sulfur amino acids, specifically cysteine [14]. Cysteine is an important component in the synthesis of glutathione, which is crucial for host defense against oxidative stress [15,16]. Wu (2017) found that, in chickens, Met can increase the relative weight of both the bursa of Fabricius and the spleen [17]. This suggests that Met is involved in the regulation of antioxidant capacity and the immune system in poultry; however, further studies in this area are required.

This the first study to investigate the effects of dietary Met on immune function and antioxidant function in partridge shank broilers with Eimeria infection. Our results may be considered partial evidence for the role of dietary Met in regulating growth performance and intestinal immune function, and for its possible mechanisms in partridge shank broilers. In addition, the optimal level of Met supplementation for partridge shank broilers is also reported for the first time in this paper. The findings set out below may provide a reference for better formulating the feed of commercially reared poultry in the future. 

## 2. Materials and Methods

### 2.1. Feed and Management

The basic diet is shown in Table 1. All kinds of nutrients (Met and sulfur-containing amino acids in the basal diet can meet 70% the of broiler requirements) meet the optimal growth needs of partridge shank broilers according to the Chinese broiler standard (NY/T33-2004). Broilers were fed for 21 days with five Met levels (0.33%, 0.39%, 0.45%, 0.51%, or 0.57%), using crystalline DL-methionine (effective content ≥ 99%, provided by Degussa, Berlin, Germany). 

Partridge shank broilers were purchased from a Chinese hatchery. A total of 1800 1-day-old chicks (about 40 g) were randomly and evenly allocated evenly into 10 treatment groups under a 2 × 5 arrangement of two anticoccidial programs (anticoccidial drug or coccidia vaccine) and 5 dietary Met levels (0.33%, 0.39%, 0.45%, 0.51% or 0.57%) as shown in Table 2. A total of 6 replicates of each treatment were carried out, with 30 chickens per group, in a pen area of dimensions 2 m × 2 m = 4 m^2^. The experimental broilers were given free access to feed and water in a constantly illuminated and temperature-controlled house. The temperature of the room was maintained at 30 to 33 °C for the first day and then reduced by 0.5 °C per day. Humidity was maintained at 60–75%. Birds were immunized according to the immunization program of yellow-feathered broilers described in [18]. Vaccinations against Marek’s disease and bursal disease were given on the 1st day; against Newcastle disease, infectious bronchitis and H9 avian influenza vaccine on the 7th day; and against H5 avian influenza vaccine (by means of wing muscle injection) on the 15th day. Birds in the coccidia vaccine groups were immunized with Schering Plough 7-valent coccidia vaccine mixed with feed on the 3rd day. Immunization cycles consisted of 7 days. Broilers were immunized by means of oocysts in bedding for the second and third immunizations. On the 17th day, feces were collected to observe the immune activity of broilers and to determine whether immunization had been successful. The anticoccidial drug group was fed 12% methyl salinomycin as an anticoccidiosis drug (Shandong Shen-gli Co., Ltd., Jinan, China,). The trial period was 21 days. The procedures used in the study and the use of all broilers were approved by the Laboratory Animal Welfare and Ethics Committee of Xichang University.

### 2.2. Sample Collection

To determine the number of coccidia oocysts, on the 17th day of the experiment, fresh feces were collected from the enclosure (40 samples per treatment, in order to better cover each chicken). Samples were collected from the small intestine or the cecum, depending upon the shape of the feces. The ratio of small intestine to cecal samples was 4:1, and samples were then mixed to measure the number of coccidia oocysts in feces and thus determine whether the coccidia vaccine had been successful.

On day 21 of the experiment, blood samples (about 5 mL each) were taken from wing veins and centrifuged at 3000× *g* for 15 min at 4 °C to separate the serum of broilers (an average body weight was calculated for each replicate for serum-determination purposes) so that biochemical indexes and antioxidant capacity indexes could be tested. In each treatment group, a single broiler of close-to-average weight was repeatedly selected; 10 mL of blood was then collected from a wing vein and added into a vial containing 1.5 mL EDTA–Na2 anticoagulant (Shanghai Yubo Biotechnology Co., Ltd., Shanghai, China). The number of T lymphocytes was determined by flow cytometry (BD FACSCanto II, San Jose, CA, USA).

One broiler close to the average weight was selected from each replicate. After slaughter, the thymus, bursa of Fabricius, and spleen were taken and weighed to calculate the relative index of immune organs (=immune organ weight/body weight). The intestinal tract was examined, and the intestinal lesion score was recorded.

Ileum (about 5 cm in ileal central section) was also extracted from slaughtered broilers, placed in a dish, cut lengthways, and scraped. Samples from the same repetition were collected in the same EP tube (Shanghai Yubo Biotechnology Co., Ltd., Shanghai, China). and stored in liquid nitrogen at −80 °C for testing.

### 2.3. Performance and Fecal Oocyst Counts

Average daily gain (ADG), average daily feed intake (ADFI), and feed-to-gain ratio (F/G) were measured on day 21. Chickens were required to fast for 8 h. Average final weight, average daily feed intake (ADFI), average daily gain (ADG), and feed weight ratio (F/G) were then determined for each stage for each repeat. Excreta were observed daily after the oocyst challenge, and oocyst counts were determined in excreta samples taken from each pen at 17 days after challenge. Oocyst populations were enumerated by microscopy according to the procedure of Long et al. (1976) [19]. The final oocyst counts were averages of those taken 17 days after challenge.

### 2.4. Histological Examination

Cecum lesions were examined using a Nikon light microscope (Nikon Eclipse 80i, Nikon Corp., Tokyo, Japan). Lesion scores were determined and recorded according to the method of Johnson et al. (1970) [20], with scores ranging from 0 (no gross lesion) to 4 (most severe gross lesion).

### 2.5. Antioxidant Substance and Immune Substances

Enzyme-linked immunosorbent assay (ELISA) sIgA levels in cecal lumina were examined using the commercially available ELISA kit (Sigma, St. Louis, MO, USA) following the manufacturer’s protocols. Levels of malondialdehyde (MDA), glutathione peroxidase (GSH-Px), total antioxidant capacity (T-AOC), T-protein, albumin, and urea nitrogen were measured using reagent kits (Nanjing Jiancheng Bioengineering, Nanjing, China). Concentrations of total protein, albumin and urea nitrogen were measured by ultraviolet spectrophotometer, also using reagent kits (Nanjing Jiancheng Bioengineering, Nanjing, China).

### 2.6. Statistical Analysis

The effects of supplemental Met upon partridge shank broilers subject to coccidia treatment were analyzed using SAS software (SAS Institute, 1998, Cary, NC, USA). Data were checked for normal distribution and homogeneity of variance using the Shapiro–Wilk test and Levene’s test. Data were further analyzed as a 5 × 2 (5 Met levels × 2 types of coccidia treatment) factorial arrangement of treatments by 2-way ANOVA with a model that included the main effects of Met levels and coccidia treatments, as well as their interaction. When an effect was significant (*p* < 0.05), means were compared by Duncan’s multiple range test to determine specific differences among means. Linear and quadratic regression contrasts were included in the analysis to determine the response of animals to increasing levels of dietary Met and different types of coccidia treatment. Met requirements were estimated using a broken-line model, and results were reported as mean ± standard deviation [21]. Unless otherwise noted, all statements of significance are based on testing at *p* < 0.05.

## 3. Results and Analysis

### 3.1. Performance

As shown in Table 3, values of final weight, ADFI, and ADG were all higher for chickens in the anticoccidial drug group, compared with birds that received the coccidia vaccine (*p* < 0.001). A lower F/G value was also recorded for birds in the anticoccidial drug group, compared with those in the coccidia vaccine group (*p* < 0.001). Final weight and ADG were both upregulated in anticoccidial drug control birds as dietary Met supplement was added, up to a level of 0.45 percent (interaction, *p* < 0.05). Above this level, the trend line was flat (linear and quadratic effect, *p* < 0.001). A Met supplementation level of 0.45% produced the highest ADFI (quadratic effect, *p* < 0.001), while supplementation at 0.57% resulted in the lowest F/G value (linear, quadratic, and cubic effect, *p* < 0.05).

### 3.2. OPG and Intestinal Lescion Score

The number of fecal coccidia oocysts was higher in the coccidia vaccine group com-pared with the anticoccidial drug group (*p* < 0.001), as shown in Table 4. The OPG linearly increased (*p* < 0.001) with dietary Met supplementation under both coccidia treatment (interaction, *p* < 0.05). The anticoccidial drug group had lower intestinal lesion scores compared with the coccidia vaccine group (*p* < 0.05). Intestinal lesion scores increased with dietary Met supplementation under anticoccidial drug treatment (linear and quadratic effect, *p* < 0.05).

### 3.3. Serum Biochemical Indexes

The activities of GSH-Px and T-AOC in broiler serum were lower in vaccinated birds compared with the anticoccidial drug group (*p* < 0.001), as shown in Table 5. However, neither of the two coccidia treatment types affected the content of MDA, T-protein, albumin, or urea nitrogen in broiler serum (*p* > 0.05). GSH-Px was upregulated with dietary Met supplementation under both types of coccidia treatment (linear and quadratic effect, *p* < 0.001) and interaction effects were presented (*p* < 0.001). T-AOC and albumin content in broiler serum were upregulated with dietary Met supplementation under both types of coccidia treatment (linear effect, *p* < 0.01). T-protein content was also upregulated with dietary Met supplementation (linear effect, *p* < 0.001) under both types of treatment and inter-action effects were again presented (*p* = 0.041). Met supplementation levels of 0.33 and 0.57% produced the highest urea nitrogen levels in broiler serum for both types of coccidia treatment (quadratic effect, *p* < 0.001). However, Met supplementation levels did not affect the content of MDA in broiler serum (*p* > 0.05).

### 3.4. Immune Function

As shown in Table 6, the bursa of Fabricius index was higher in broilers that received the anticoccidial drug, compared with birds in the coccidia vaccine group (*p* < 0.001). However, neither type of treatment affected the spleen index or thymus index of broilers (*p* > 0.05). The bursa of Fabricius index was upregulated with dietary Met supplementation in broilers under both types of coccidia treatment (linear and quadratic effect, *p* < 0.05). The bursa of Fabricius index was upregulated in broilers with dietary Met supplementation under coccidia treatment (quadratic effect, *p* < 0.001), and interaction effects were presented (*p* = 0.046). However, Met supplementation did not affect the thymus index of broilers (*p* > 0.05).

The coccidia vaccine increased the level of CD^4+^ and the ratio of CD^4+^/CD^8+^ in broiler blood, compared with birds in the anticoccidial drug group (*p* < 0.001), as set out in Table 7. However, neither type of treatment affected the level of CD^8+^ or the content of sIgA in broiler blood (*p* > 0.05). The blood content of sIgA was upregulated with dietary Met supplement under both types of coccidia treatment (linear and quadratic effect, *p* < 0.001). However, aside from this main effect, Met supplementation did not affect the ratio of CD^4+^ and CD^8+^ or the value of CD^4+^/CD^8+^ (*p* > 0.05). The results of multiple comparisons revealed that appropriate levels of Met supplementation increased the level of CD^4+^ (rather than CD^8+^) and the ratio of CD^4^+/CD^8+^ in broilers under both types of coccidia treatment.

## 4. Discussion

In this research, we found that the ADFI, ADG, final weight, and F/G were raised with dietary Met supplements added to 0.45%; after this ADG and final weight trend plain, ADFI and F/G slowed down under both coccidia treatment. These results indicate that low levels of dietary Met cannot meet the production needs of broilers. Performance can be improved with Met supplementation, but excessive Met supplementation might have adverse effects. In the present study, the anticoccidial drug group had higher final weights, higher ADFI and ADG values, and lower F/G values compared with the coccidia vaccine group, indicating that the coccidia vaccine reduces the performance of broilers in comparison with birds that receive anticoccidial drugs. It may be that the coccidia vaccine affects the intestinal structure of broilers and reduces nutrient absorption and utilization rates. To consider this matter more fully, we carried out further research and found that OPG and intestinal injury scores were higher in coccidia-vaccinated broilers than in birds which received anticoccidial drug treatment. These results indicated the success of coccidia vaccine immunization and the negative effects of coccidia vaccine on the intestinal structure and function of broilers. In addition, there is an interaction effect between the coccidia treatment and Met level on final weight and ADG, which indicated the different regulation of Met on production performance between two coccidia treatment in the partridge shank. In addition, we found an interesting result: the OPG linearly increased with dietary Met supplementation under coccidia treatment. The effect of Met on coccidia needs further study.

Met is involved in the regulation of physiological functions through regulating the REDOX system and protein synthesis [22]. In the serum of partridge shank broilers, the activities of GSH-Px (rather than MDA and T-AOC) increased as Met supplementation levels rose. This may suggest that Met participates in the regulation of the antioxidant system and that GSH-Px activity might be more sensitive to levels of Met supplementation in the diets of broilers under coccidia treatment. The involvement of dietary Met in the regulation of GSH-Px activity might also be related to cysteine. Met plays a significant role in the syn-thesis of cysteine [23], which is an important component of GSH [24]. In addition, in this study, the coccidia vaccine resulted in lower GSH-Px activity and lower T-AOC levels, causing the interaction effect of GSH-Px activity. Some previous studies have found that free radical content and antioxidant enzyme activity in broilers decreased after coccidia infection [25,26].

In broilers, Met is mainly used in protein synthesis [27]. In this experiment, dietary Met deficiency significantly reduced total protein and albumin content in the serum of birds under both types of coccidia treatment program. Blood urea nitrogen is wholly a product of amino acid metabolism and can reflect the metabolic status of amino acids in the body [28]. In this experiment, levels of blood urea nitrogen concentration exhibited a quadratic curve with changes in dietary Met supplementation, indicating that supplementation levels that are too high or too low can both affect the utilization of amino acids in broilers.

When coccidia invade the intestinal epithelial cells of broilers, specific and nonspecific immune responses are triggered [29], especially mucosal immunity and development in the immune organs [3]. The bursa of Fabricius is an important immune organ in broiler chickens, being the main organ involved in the differentiation and maturation of B lymphocytes that are closely involved in humoral immunity. In this study, broilers that received the anticoccidial drug had higher Fabricius index values, indicating that anticoccidial drug treatment is conducive to the bursa of Fabricius development. Dietary Met also produced a significant effect on the bursa of Fabricius and thymus (rather than spleen) index values of broilers in this study. Rochell (2015) reported that when Met deficiency reduced the number of thymus cells in broilers, resulting in thymus damage, the cellular immunity of birds was ultimately affected [30]. In this study, we found an interaction between coccidia treatment and Met levels, which may be related to the cellular immune response to coccidioid infection, and the fact that the thymus is the main organ for T lymphocyte differentiation and maturation. The number of CD^4+^T lymphocytes and the ratio of CD^4+^/CD^8+^ were both higher in the blood of broilers that received coccidia vaccine treatment. During coccidia immunity, CD^4+^T lymphocytes function as helper T lymphocytes, while CD^8+^ T lymphocytes function as effector T lymphocytes [10]. The CD^4+^/CD^8+^ ratio is usually maintained in a range of 1 to 2, and any increase or decrease in the CD^4+^/CD^8+^ ratio indicates immunosuppression or immunoenhancement [31].

The coccidia vaccine is based on cellular immunity. In this study, the number of CD4+ and CD^8+^T lymphocytes in vaccinated broilers was higher than in drug-treated birds. Met supplementation improved the level of CD^4+^ (rather than CD^8+^) and the ratio of CD^4+^/CD^8+^ in birds of both treatment groups. An adequate level of Met plays an important role in the synthesis of small peptides and proteins required by the immune system [32]. On the other hand, Met metabolites such as polyamine, GSH, Tau, and Hcy have important regulatory roles in specific and nonspecific immune functions. The function of T lymphocytes de-pends on the concentration of GSH. It has been reported that T lymphocyte function may be affected by deficiency in sulfur-containing amino acids [32]. Wu et al. (2012) also showed that an adequate level of Met can significantly induce the percentage of CD4+ and CD^8+^T lymphocytes, and the ratio of CD^4+^ to CD^8+^ [8]. However, in this study, dietary Met did not affect levels of CD^4+^ and CD^8+^, and this finding was in line with previous research carried out in our laboratory [18]. In this study, we found that 0.45% Met supplementation produced the highest CD^4+^ cell percentages and CD^4+^/CD^8+^ ratios in vaccinated chickens. To our surprise, we found a contrary result with respect to CD^8+^. This may be related to the longer feed cycle which may activate the fecal system more effectively. IgA is the main antibody isoform present in bodily secretions and acts as a first line of defense against pathogens which invade and colonize the intestinal mucosa. It has previously been shown that the IgA titers in the intestinal lumen are mirrored by serum IgA titers [33]. This study found that appropriate levels of Met can increase the IgA content in broiler blood, indicating that improvements in immune function brought about by Met are related to IgA.

The above data obviously indicate that an appropriate level of dietary Met improves performance, antioxidant capacity, and immune function (IgA content in cecal lumina) in partridge shank broilers. According to the quadratic regression equation model (shown in Table 8), the optimal level of digestible Met for ADG (1–21 days) was estimated to be 0.418% and 0.444% of diet, respectively, for each type of coccidia treatment. With respect to F/G (1–21 days), the optimal level of digestible Met was estimated to be 0.451% and 0.455% of diet, respectively, for each type of coccidia treatment. With sIgA of ileum mucosa as evaluation index, the requirements of total Met and digestible Met for broilers aged from 1 to 21 days under two coccidia treatment scheme were 0.486% and 0.451%, respectively. This result might indicate that partridge shank broiler might require more Met and Met metabolites to resistance to coccidia infection under coccidia vaccine.

## 5. Conclusions

In this study, we found that optimal levels of dietary Met improve the production of partridge shank broilers. This might be the result of changes in immune function (IgA in cecal lumina) and antioxidant capacity. We also found that coccidia vaccine reduces the performance of broilers, compared with birds which receive anticoccidial drugs for the first time. Optimal levels of digestible Met in terms of production performance (ADG and F/G) and immune function (sIgA in ileum mucosa) in partridge shank broilers (1–21 days) were found to be 0.418%, 0.451%, and 0.451% of diet, respectively, when birds received anticoccidial drug treatment. For birds which received the coccidia vaccine, the equivalent figures were 0.444%, 0.455%, and 0.452 %, as shown in Table 8. In conclusion, the present study reveals the optimal dietary Met requirements for partridge shank broilers (1–21 days) which receive either anticoccidial drug treatment or a coccidia vaccine. Our results may provide a reference for better formulating the feed of commercially reared poultry in the future.

## Figures and Tables

**Table 1 animals-13-00613-t001:** Composition and nutrient levels of basal diet (air-dry basis,%).

Ingredient	1~21 d (%)
Corn	55.14
Wheat	5.00
High protein soybean meal (43%)	35.00
Soybean oil	1.30
Calcium hydrophosphate	1.85
Calcium carbonate	1.16
Choline chloride	0.10
Nacl	0.32
L-lysine (98%)	0.03
Premix ^1^	0.10
Total	100.00
Nutritional level ^2^	
TMEn (MJ/kg)	12.14
Crude protein	21.01
Lysine	1.07
Methionine	0.33
Sulphur amino acids	0.68
Threonine	0.76
Ca	1.00
Total *p*	0.69

^1^ Provided following per kg of premix: Vitamin A 8000 IU, Vitamin D3 2000 IU, Vitamin E 15U, Vitamin K3 1.5 mg, Vitamin B1 2 mg, Vitamin B2 5 mg, Vitamin B6 5 mg, Vitamin B12 0.012 mg, folic acid 0.7 mg, niacin 40 mg, Panpantothenate 12 mg, biotin 0.2 mg, Cu 10 mg (as copper sulfate), Fe 90 mg (As ferrous sulfate), Mn 100 mg (as manganese sulfate), Zn 100 mg (as zinc sulfate), Se 0.3 mg (as sodium selenite), I 0.5 mg (as calcium iodate).^2^ Nitrogen corrected true metabolizable energy (TMEn) was calculated, and other nutritional indexes were measured. Among the nutrient levels, TMEn was calculated, and other nutrient levels were measured.

**Table 2 animals-13-00613-t002:** Experimental design.

Treatment	Coccidia Treatment	Met Level (%)
		1–21 d
1	Anti-coccidial drug	0.33
2		0.39
3		0.45
4		0.51
5		0.57
6	Coccidia vaccine	0.33
7		0.39
8		0.45
9		0.51
10		0.57

Anti-coccidial drug (narasin, 60 mg/kg diet) was added to diets in medicated groups, and an anticoccidial 7-valent live vaccine was inoculated in vaccinated groups.

**Table 3 animals-13-00613-t003:** Effect of different treatments on the growth performance of 1~21 d partridge shank broilers.

Coccidia Treatment	Met Level (%)	Initial Weight	Final Weight	ADFI	ADG	F/G
		(g)	(g)	(g)	(g)	
Anti-coccidial drug	0.33	32.54 ± 0.34	386 ± 10.52 ^e^	29.87 ± 0.60 ^abcde^	17.67±0.52 ^e^	1.69 ± 0.04 ^b^
	0.39	32.63 ± 0.26	412 ± 5.08 ^bc^	30.38 ± 0.75 ^abc^	18.93±0.27 ^bc^	1.61 ± 0.03 ^def^
	0.45	32.58 ± 0.35	425 ± 6.98 ^a^	30.88 ± 0.46 ^a^	19.62±0.37 ^a^	1.58 ± 0.03 ^f^
	0.51	32.63 ± 0.38	417 ± 4.58 ^ab^	30.57 ± 0.33 ^ab^	19.20±0.2 ^ab^	1.59 ± 0.03 ^df^
	0.57	32.67 ± 0.31	409 ± 14.87 ^bc^	29.45 ± 0.43 ^cde^	18.83±0.29 ^bc^	1.60 ± 0.02 ^f^
Coccidia vaccine	0.33	32.55 ± 0.34	364 ± 9.88 ^f^	28.92 ± 0.64 ^e^	16.58±0.52 ^f^	1.75 ± 0.05 ^a^
	0.39	32.63 ± 0.26	383 ± 12.78 ^e^	29.27 ± 1.33 ^de^	17.57 ± 0.63 ^e^	1.67 ± 0.04 ^bc^
	0.45	32.59 ± 0.35	401 ± 3.81 ^cd^	30.28 ± 1.08 ^abcd^	18.42 ± 0.20 ^cd^	1.65 ± 0.06 ^bcd^
	0.51	32.63 ± 0.25	401 ± 13.19 ^cd^	30.02 ± 0.93 ^abcd^	18.44 ± 0.62 ^cd^	1.63±0.04 ^cde^
	0.57	32.67 ± 0.36	399 ± 19.16 ^d^	29.73 ± 0.75 ^bcde^	18.22 ± 0.60 ^d^	1.64 ± 0.04 ^cde^
Main effect						
Coccidia treatment	Anti-coccidial drug	32.63 ± 0.30	409.61 ± 14.87 ^a^	30.23 ± 0.72 ^a^	18.85 ± 0.74 ^a^	1.61 ± 0.06 ^b^
	Coccidia vaccine	32.63 ± 0.30	390.57 ± 18.22 ^b^	29.51 ± 1.03 ^b^	17.91 ± 0.90 ^b^	1.65 ± 0.08 ^a^
Met level (%)	0.33	32.55 ± 0.32	374.78 ± 14.97 ^c^	29.39 ± 0.77 ^c^	17.13 ± 0.75 ^c^	1.72 ± 0.05 ^a^
	0.39	32.6 ± 0.25	397.38 ± 17.38 ^b^	29.83 ± 1.18 ^bc^	18.25 ± 0.85 ^b^	1.64 ± 0.05 ^b^
	0.45	32.65 ± 0.34	413.08 ± 13.69 ^a^	30.58 ± 0.85 ^a^	19.02 ± 0.69 ^a^	1.61 ± 0.06 ^b^
	0.51	32.65 ± 0.25	408.76 ± 12.49 ^a^	30.29 ± 0.72 ^ab^	18.81 ± 0.60 ^a^	1.61 ± 0.04 ^b^
	0.57	32.58 ± 0.36	406.46 ± 8.15 ^a^	29.25 ± 0.46 ^c^	18.69 ± 0.40 ^a^	1.57 ± 0.04 ^c^
*p*-Value						
Coccidia treatment		1	<0.001	<0.001	<0.001	<0.001
Met		0.867	<0.001	<0.001	<0.001	<0.001
Coccidia treatment × Met		1	0.027	0.765	0.033	0.279
Linear effect of Met		0.37	<0.001	0.793	<0.001	<0.002
Quadratic effect of Met		0.92	<0.001	<0.001	<0.001	0.031
Cubic effect of Met		0.65	0.276	0.128	0.285	0.009

In the same column, values with different small-letter superscripts mean significant difference (*p* < 0.05). Met: Methionine; ADFI: Average feed intake; ADG: Average daily growth; F/G: Feed/gain.

**Table 4 animals-13-00613-t004:** Effect of different treatments on the OPG and intestinal lesion score of 1~21 d partridge shank broilers.

Coccidia Treatment	Met Level (%)	OPG (1 × 10^3^)	Intestinal Lesion Score
Anti-coccidial drug	0.33	0.67 ± 0.82 ^d^	0.22 ± 0.12 ^c^
	0.39	0.50 ± 0.84 ^d^	0.18 ± 0.10 ^bc^
	0.45	0.33 ± 0.52 ^d^	0.12 ± 0.09 ^c^
	0.51	0.17 ± 0.41 ^d^	0.13 ± 0.05 ^bc^
	0.57	0.17 ± 0.41 ^d^	0.17 ± 0.05 ^bc^
Coccidia vaccine	0.33	7.50 ± 0.84 ^c^	0.30 ± 0.17 ^a^
	0.39	9.33 ± 1.75 ^b^	0.23 ± 0.05 ^ab^
	0.45	10.34 ± 1.75 ^ab^	0.18 ± 0.05 ^bc^
	0.51	10.50 ± 1.64 ^ab^	0.16 ± 0.05 ^bc^
	0.57	11.50 ± 1.37 ^a^	0.18 ± 0.10 ^bc^
Main effect			
Coccidia treatment	Anti-coccidial drug	0.37 ± 0.61 ^b^	0.16 ± 0.08 ^b^
	Coccidia vaccine	9.83 ± 1.97 ^a^	0.21 ± 0.10 ^a^
Met level (%)	0.33	4.08 ± 3.65 ^b^	0.26 ± 0.14 ^a^
	0.39	4.92 ± 4.80 ^ab^	0.21 ± 0.08 ^ab^
	0.45	5.33 ± 5.37 ^a^	0.15 ± 0.07 ^b^
	0.51	5.33 ± 5.52 ^a^	0.15 ± 0.05 ^b^
	0.57	5.83 ± 6.00 ^a^	0.18 ± 0.08 ^b^
*p*-Value			
Coccidia treatment		<0.001	0.039
Met level		0.009	0.016
Coccidia treatment × Met level		<0.001	0.910
Linear effect of Met level		<0.001	0.007
Quadratic effect of Met level		0.390	0.030

In the same column, values with no letter or the same letter superscripts mean no significant difference (*p* > 0.05), while with different small-letter superscripts mean significant difference (*p* < 0.05). Met: Methionine; OPG: Oocysts per gram feces.

**Table 5 animals-13-00613-t005:** Effect of different treatments on serum biochemical indexes of 1~21 d partridge shank broilers.

Coccidia Treatment	Met Level (%)	GSH-Px	MDA	T-AOC	T-Protein(g/L)	Albumin	Urea Nitrogen (mmol/L)
		(U/mL)	(nmol/mL)	(U/mL)		(g/L)	
Anti-coccidial drug	0.33	1035.32 ± 22.91 ^g^	7.32 ± 1.27	8.42 ± 1.36 ^ab^	28.62 ± 2.06 ^c^	12.78 ± 1.49 ^abc^	0.93 ± 0.13 ^a^
	0.39	1147.88 ± 22.29 ^e^	6.88 ± 1.43	8.57 ± 1.34 ^ab^	31.87 ± 2.56 ^b^	13.25 ± 1.22 ^abc^	0.69 ± 0.10 ^c^
	0.45	1235.83 ± 28.75 ^bc^	6.55 ± 1.61	8.93 ± 1.43 ^ab^	32.33 ± 2.77 ^b^	13.26 ±1.85 ^abc^	0.65 ± 0.07 ^c^
	0.51	1313.90 ± 37.35 ^a^	6.68 ± 1.88	9.58 ± 1.28 ^a^	33.31 ± 2.64 ^ab^	13.78 ± 1.23 ^abc^	0.73 ± 0.11 ^bc^
	0.57	1271.00 ± 47.29 ^b^	7.00 ± 0.92	9.20 ± 1.30 ^a^	34.90 ± 3.88 ^ab^	14.38 ± 1.90 ^a^	0.90 ± 0.13 ^a^
Coccidia vaccine	0.33	870.22 ± 24.09 ^h^	8.27 ± 1.64	7.35 ± 1.10 ^b^	26.78 ± 2.39 ^c^	11.88 ± 1.27 ^c^	0.86 ± 0.11 ^ab^
	0.39	1088.20 ± 17.33 ^f^	7.31 ± 1.27	8.08 ± 1.32 ^ab^	27.98 ± 2.37 ^c^	12.30 ± 1.45 ^bc^	0.72 ± 0.11 ^bc^
	0.45	1192.73 ± 27.85 ^d^	6.55 ± 1.61	8.45 ± 1.09 ^ab^	32.48 ± 2.20 ^b^	13.01 ± 1.35 ^abc^	0.70 ± 0.09 ^c^
	0.51	1217.77 ± 32.40 ^cd^	6.68 ± 1.88	8.77 ± 1.10 ^ab^	35.98 ± 2.31 ^a^	13.98 ± 1.53 ^ab^	0.72 ± 0.11 ^bc^
	0.57	1217.25 ± 34.58 ^cd^	6.80 ± 1.08	8.79 ± 1.00 ^ab^	34.90 ± 2.16 ^ab^	14.13 ± 1.64 ^ab^	0.83 ± 0.14 ^b^
Main effect							
Coccidia treatment	anti-coccidial drug	1200.79 ± 105.41 ^a^	6.89 ± 1.38	8.95 ± 1.32 ^a^	32.21 ± 3.38	13.49 ± 1.56	0.78 ± 0.15
	Coccidia vaccine	1117.23 ± 137.16 ^b^	7.12 ± 1.55	8.28 ± 1.19 ^b^	31.63 ± 4.29	13.06 ± 1.63	0.75 ± 0.12
Met level (%)	0.33	952.77 ± 89.09 ^d^	7.79 ± 1.48	7.87 ± 1.32	27.72 ± 2.33 ^d^	12.33 ± 1.40 ^c^	0.89 ± 0.12 ^a^
	0.39	1118.04 ± 36.52 ^c^	7.10 ± 1.31	8.33 ± 1.30	29.93 ± 3.10 ^c^	12.78 ± 1.37 ^bc^	0.71 ± 0.10 ^b^
	0.45	1214.28 ± 35.14 ^b^	6.55 ± 1.54	8.70 ± 1.24	32.41 ± 2.38 ^b^	13.14 ± 1.55 ^abc^	0.68 ± 0.08 ^b^
	0.51	1265.83 ± 60.26 ^a^	6.68 ± 1.80	9.18 ± 1.22	34.65 ± 2.74 ^a^	13.88 ± 1.32 ^ab^	0.73 ± 0.11 ^b^
	0.57	1244.13 ± 48.46 ^a^	6.90 ± 0.96	9.00 ± 1.13	34.90 ± 3.00 ^a^	14.26 ± 1.70 ^a^	0.83 ± 0.15 ^a^
*p*-Value							
Coccidia treatment		<0.001	0.542	0.043	0.388	0.276	0.313
Met		<0.001	0.291	0.086	<0.001	0.018	<0.001
Coccidia treatment × Met Level		<0.001	0.885	0.950	0.041	0.868	0.225
Linear effect of Met		<0.001	0.113	0.008	<0.001	<0.001	0.301
Quadratic effect of Met		<0.001	0.127	0.388	0.139	0.883	<0.001

In the same column, values with different small letter superscripts mean significant difference (*p* < 0.05). Met: Methionine; GSH-Px: Glutathione peroxidase; MDA: Malonaldehyde; T-AOC: Total antioxidant capacity.

**Table 6 animals-13-00613-t006:** Effect of different treatments on immune organ indexes of 1~21 d partridge shank broilers.

Coccidia Treatment	Met Level (%)	Spleen Index	Bursa of Fabricius Index	Thymus Index
Anti-coccidial drug	0.33	0.10 ± 0.01 ^ab^	0.24 ± 0.04 ^d^	0.31 ± 0.03 ^d^
	0.39	0.11 ± 0.03 ^ab^	0.35 ± 0.05 ^bc^	0.47 ± 0.11 ^ab^
	0.45	0.12 ± 0.03 ^ab^	0.44 ± 0.11 ^a^	0.34 ± 0.08 ^d^
	0.51	0.09 ± 0.01 ^b^	0.35 ± 0.04 ^bc^	0.46 ± 0.07 ^ab^
	0.57	0.15 ± 0.10 ^a^	0.40 ± 0.08 ^ab^	0.39 ± 0.07 ^bcd^
Coccidia vaccine	0.33	0.09 ± 0.04 ^b^	0.27 ± 0.10 ^d^	0.38 ± 0.06 ^cd^
	0.39	0.14 ± 0.01 ^ab^	0.32 ± 0.07 ^bcd^	0.48 ± 0.06 ^a^
	0.45	0.11 ± 0.02 ^ab^	0.31 ± 0.03 ^cd^	0.43 ± 0.05 ^abc^
	0.51	0.09 ± 0.01 ^b^	0.33 ± 0.05 ^bc^	0.46 ± 0.07 ^ab^
	0.57	0.11 ± 0.03 ^ab^	0.33 ± 0.06 ^bcd^	0.32 ± 0.03 ^d^
Main effect				
Coccidia treatment	anti-coccidial drug	0.12 ± 0.05	0.36 ± 0.09 ^a^	0.39 ± 0.10
	Coccidia vaccine	0.11 ± 0.03	0.31 ± 0.07 ^b^	0.41 ± 0.08
Met level (%)	0.33	0.10 ± 0.03	0.26 ± 0.07 ^b^	0.34 ± 0.06 ^b^
	0.39	0.13 ± 0.03	0.33 ± 0.06 ^a^	0.47 ± 0.08 ^a^
	0.45	0.11 ± 0.02	0.37 ± 0.10 ^a^	0.38 ± 0.08 ^b^
	0.51	0.09 ± 0.01	0.34 ± 0.05 ^a^	0.46 ± 0.07 ^a^
	0.57	0.13 ± 0.08	0.37 ± 0.08 ^a^	0.35 ± 0.06 ^b^
*p*-Value				
Coccidia treatment		0.626	0.012	0.250
Met		0.062	0.001	<0.001
Coccidia treatment × Met		0.269	0.080	0.046
Linear effect of Met		0.410	<0.001	0.871
Quadratic effect of Met		0.727	0.018	<0.001

In the same column, values with different small letter superscripts mean significant difference (*p* < 0.05).

**Table 7 animals-13-00613-t007:** Effect of different treatments on immunologic function of 1~21 d partridge shank broilers.

Coccidia Treatment	Met Level (%)	CD4^+^ (%)	CD8^+^ (%)	CD4^+^/CD8^+^	sIgA (mg/mL)
Anti-coccidial drug	0.33	11.82 ± 1.10 ^bcd^	8.40 ± 0.80	1.42 ± 0.17 ^ab^	0.80 ± 0.12 ^d^
	0.39	12.19 ± 1.65 ^abc^	8.40 ± 0.69	1.45 ± 0.19 ^ab^	1.03 ± 0.10 ^bc^
	0.45	12.03 ± 1.54 ^abc^	8.53 ± 0.54	1.41 ± 0.11 ^ab^	1.16 ± 0.12 ^ab^
	0.51	11.51 ± 1.35 ^cd^	8.20 ± 0.72	1.41 ± 0.18 ^ab^	1.19 ± 0.10 ^a^
	0.57	10.15 ± 1.11 ^d^	8.07 ± 0.63	1.27 ± 0.20 ^b^	1.17 ± 0.15 ^ab^
Coccidia vaccine	0.33	12.67 ± 1.32 ^abc^	8.65 ± 1.11	1.47 ± 0.14 ^ab^	0.77 ± 0.12 ^d^
	0.39	13.37 ± 1.60 ^abc^	8.80 ± 0.93	1.53 ± 0.15 ^a^	1.00 ± 0.10 ^c^
	0.45	13.89 ± 1.87 ^a^	8.75 ± 0.89	1.59 ± 0.20 ^a^	1.13 ± 0.12 ^abc^
	0.51	13.66 ± 1.64 ^ab^	8.55 ± 0.62	1.60 ± 0.15 ^a^	1.15 ± 0.10 ^ab^
	0.57	13.27 ± 1.37 ^abc^	8.57 ± 0.71	1.55 ± 0.12 ^a^	1.14 ± 0.15 ^abc^
Main effect					
Coccidia treatment	anti-coccidial drug	11.54 ± 1.47 ^b^	8.32 ± 0.65	1.39 ± 0.17 ^b^	1.07 ± 0.19
	Coccidia vaccine	13.37 ± 1.52 ^a^	8.66 ± 0.81	1.55 ± 0.15 ^a^	1.04 ± 0.18
Met level (%)	0.33	12.24 ± 1.24	8.53 ± 0.93	1.45 ± 0.15	0.78 ± 0.11 ^c^
	0.39	12.78 ± 1.67	8.60 ± 0.81	1.49 ± 0.17	1.01 ± 0.10 ^b^
	0.45	12.95 ± 1.90	8.64 ± 0.71	1.50 ± 0.18	1.15 ± 0.11 ^a^
	0.51	12.59 ± 1.82	8.38 ± 0.67	1.51 ± 0.19	1.17 ± 0.10 ^a^
	0.57	11.71 ± 2.02	8.32 ± 0.69	1.41 ± 0.22	1.15 ± 0.14 ^a^
*p*-Value					
Coccidia treatment		<0.001	0.095	0.001	0.309
Met		0.275	0.819	0.583	<0.001
Coccidia treatment × Met		0.372	0.991	0.446	1.000
Linear effect of Met		0.360	0.376	0.721	<0.001
Quadratic effect of Met		0.041	0.506	0.126	<0.001

In the same column, values with different small letter superscripts mean significant difference (*p* < 0.05). Met: Methionine; CD^4+^: CD^4+^T lymphocyte; CD^8+^: CD^8+^T lymphocyte; sIgA: secretory immuoglobulin.

**Table 8 animals-13-00613-t008:** The total Met and digestible Met requirements of 1~21 d partridge shank broiler by quadratic regression model estimation.

Evaluation Indicators	Coccidia Treatment	Quadratic Regression Equation	*p*-Value	Total Met Requirement (%)	Digestible Met Requirement (%)
ADG	Anti-coccidial drug	y = −86.083x^2^ + 82.088x − 0.015, R^2^ = 0.77	<0.001	0.453	0.418
	Coccidia vaccine	y = −64.153x^2^ + 64.627x + 2.214, R^2^ = 0.66	<0.001	0.479	0.444
F/G	Anti-coccidial drug	y =3.594x^2^ − 3.679x + 2.508, R^2^ = 0.63	<0.001	0.486	0.451
	Coccidia vaccine	y =3.340x^2^ − 3.453x + 2.518, R^2^ = 0.51	<0.001	0.49	0.455
The ileum mucosa sIgA	Anti-coccidial drug	y = −12.142x^2^ +12.44x − 1.9336, R^2^ = 0.65	<0.001	0.486	0.451
	Coccidia vaccine	y = −11.788 x^2^ +12.078x − 1.9258, R^2^ = 0.65	<0.001	0.487	0.452

## Data Availability

The data presented in this study are available upon request from the corresponding author.

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
