# Peer review of "Dietary Methionine Increased the Growth Performances and Immune Function of Partridge Shank Broilers after Challenged with Coccidia"

_animals, 2023, doi:10.3390/ani13040613_

Round 1
Reviewer 1 Report
The authors investigated the effects of methionine on production performance such as growth, immune function, and antioxidant capacity with treatment of coccidioides and coccidioides vaccine. I have several concerns as follows.
Q1, There’s no control group in the experimental design. I'm not sure whether this can be accepted for audience.
Q2. What breed do you used in your experiment? Partridge shank broilers? Does this mean Ross, Cobb, AA+ commercial line? Because the total experiment was done from 0d to 21days. But final weights are only 300~400g. If no challenge, what about the normal body weight for this breed?
Q3. I think you can replot the table with some horizontal line between coccidioides and coccidia vaccines group.
Q4. I think the F/G would not match the assumption of ANOVA. Have you test it?
Q5. Main effect of two-way ANOVA could not provide valuable information for audience. The Coccidia treatment effect can be signed on the average of these two groups. And the Met level effects has been labelled on the corresponding average value. Maybe just puzzled me. Also, it would be better if you can visualize the data.
Q6. Why not add some histological slides?
Author Response
Response to reviewer #1:
General Comment: The authors investigated the effects of methionine on production performance such as growth, immune function, and antioxidant capacity with treatment of coccidioides and coccidioides vaccine. I have several concerns as follows.
Answer:
Dear reviewer,
Thanks for your kind suggestions and comments, which are all valuable and very helpful for revising and improving our paper, as well as the important guiding significance to our researches. We have studied comments carefully and have made correction which we hope meet with approval. The detail changes are as follows.
Comment 1:
There’s no control group in the experimental design. I'm not sure whether this can be accepted for audience.
Answer: Thank you very much for your kind comment. This experiment does not design the control group without coccidiosis drugs and coccidia vaccines, which is mainly based on that coccidiosis drugs and coccidiosis vaccines are generally used in production. The incidence rate of coccidiosis is very high without coccidiosis drugs or vaccines. There is also some research about alternatives to plant extracts, but in production, it is mainly through coccidia vaccines and coccidiosis drugs. Therefore, this experiment does not set up a treatment group without coccidiosis drugs or coccidia vaccines, focuses more on the comparison of treatment effects between coccidia vaccines and coccidiosis drugs. Of course, your suggestions are very helpful, if there is a control group, the whole experimental design will be more complete, and we will seriously consider it in the future experimental design.
Comment 2:
What breed do you used in your experiment? Partridge shank broilers? Does this mean Ross, Cobb, AA+ commercial line? Because the total experiment was done from 0d to 21days. But final weights are only 300~400g. If no challenge, what about the normal body weight for this breed?
Answer: Thank you very much for your kind comment. As a Chinese local chicken, partridge shank broilers have the advantages of low ratio of feed to gain and good meat quality which does not belong to Ross, Cobb, AA+ commercial line (We mentioned it in introduction in line 37-38). Wen (2017) shows the AA broilers (FCR=1.98) had superior growth performance compared with those of the partridge shank broilers (FCR=2.34) during the 42-day study. The low ratio of feed to gain caused the final weights are only 300~400g. In our research, the F/G (Feed /gain) was 1.6 which is consistent with Zhou (2019) showed that the partridge shank broilers feed conversion ratio (FCR) is 1.52-1.6 in during the 21-day study with MOS exhibited.
Our experiment has no relevant data about no challenge, which can't give you an accurate answer. There are few studies on partridge shank broilers, especially at the age of 21 days. We speculate that it may be higher than 400g according to the study of Zhou (2019).
Reference
Wen, C.; Jiang, X. Y.; Ding, L. R.; Wang, T.; Zhou, Y. M. Effects of dietary methionine on growth performance, meat quality and oxidative status of breast muscle in fast- and slow-growing broilers. Poultry Science 2017. 96:1707–1714 http://dx.doi.org/10.3382/ps/pew432
Zhou, M.Y.; Tao Y.H.; Lai, C.H; Huang, C.X.; Zhou, Y.M.; Yong, Q. Effects of Mannanoligosaccharide Supplementation on the Growth Performance, Immunity, and Oxidative Status of Partridge Shank Chickens. Animals. 2019, 9, 817; doi:10.3390/ani9100817.
Comment 3:
I think you can replot the table with some horizontal line between coccidioides and coccidia vaccines group
Answer: Thank you very much for your kind comment. Your suggestion is very good. According to your kind suggestion, we have added horizontal line between coccidioides and coccidia vaccines group in our manuscript.
Comment 4:
I think the F/G would not match the assumption of ANOVA. Have you test it?
Answer: Thank you very much for your kind comment. We used Shapiro-Wilk model to checked the normal distribution, the P=0.2745(P>0.05) means this data conforms to a standard normal distribution which match the assumption of ANOVA. We have carried out relevant tests on the data, which are mentioned in the Materials and Methods in line 150-151.
Comment 5:
Main effect of two-way ANOVA could not provide valuable information for audience. The Coccidia treatment effect can be signed on the average of these two groups. And the Met level effects has been labelled on the corresponding average value. Maybe just puzzled me. Also, it would be better if you can visualize the data.
Answer: Thank you very much for your kind comment. Our experimental design includes two factors and ten treatments. We want to give as much data as possible through multiple comparisons, principal component analysis, interaction effects, and regression analysis. The table of data display really looks complicated but comprehensive. The figure cannot reflect all the results well, we choose the form of data display. We have added some horizontal lines to the table, which may be easier for the reader to understand.
Comment 6:
Why not add some histological slides?
Answer: Thank you very much for your kind comment. We are very sorry that we do not have histological slides, and cannot provide. It is a great pity that we know that this part of data would make the paper completer and more convincing.
Thanks again for your kind suggestion.

Reviewer 2 Report
It should be said that the authors undertook quite an interesting study, which should be of both application and practical importance in terms of optimizing the keeping of wild animals in captivity, in terms of nutrition and health. The presented results show and give an overview of the possibilities of using a specific preventive solution in the field of coccidiosis, and at the same time their impact on the productivity of birds, with the appropriate (skillful) use of The work is quite interesting, however, after reading the manuscript in detail, I came up with a few comments that, in my opinion, will allow me to improve the manuscript: In my opinion, the chapters Abstract and Summary should be revised as they present the same data Line 86-87, the scheme of the experiment is described too little. In what rooms were the birds kept. 2x5 treatments and 30 chickens and 6 repetitions - it needs to be clarified, because maybe I don't understand something, but I don't get 1080 birds here. This scheme of the experiment is not very clear, it needs to be explained in detail. In addition, cages, welfare, zoohygienic conditions, use of other pharmacological agents in nutrition, etc. should be described. Line 100, 40 samples per treatment - what does it mean? It needs to be developed Line 101, again imprecise, how should this be understood, faeces were collected from the room and separately from the birds from two sections of the digestive tract? Which feces were tested and from how many samples and from what angle? Line 98 onwards, was there approval from the ethics committee to conduct the experiment? Consideration should be given to changing some tables (No. 3, 5 and 7) because they contain too many numbers and thus become illegible. I think that taking into account the above comments will improve the manuscript and make it more readable and understandable. At the same time, I recommend the manuscript for printing.Author Response
Response to reviewer #2:
General Comment: It should be said that the authors undertook quite an interesting study, which should be of both application and practical importance in terms of optimizing the keeping of wild animals in captivity, in terms of nutrition and health. The presented results show and give an overview of the possibilities of using a specific preventive solution in the field of coccidiosis, and at the same time their impact on the productivity of birds, with the appropriate (skillful) use of The work is quite interesting, however, after reading the manuscript in detail, I came up with a few comments that, in my opinion, will allow me to improve the manuscript:
Thanks for your kind suggestions and comments, which are all valuable and very helpful for revising and improving our paper, as well as the important guiding significance to our researches. We have studied comments carefully and have made correction which we hope meet with approval. The detail changes are as follows.
Comment 1:
the chapters Abstract and Summary should be revised as they present the same data Line 86-87, the scheme of the experiment is described too little. In what rooms were the birds kept. 2x5 treatments and 30 chickens and 6 repetitions - it needs to be clarified, because maybe I don't understand something, but I don't get 1080 birds here. This scheme of the experiment is not very clear, it needs to be explained in detail.
Answer: Thank you very much for your kind comment. We are very sorry for our careless. We revised the Abstract, Material and Methods in as follows:
line 18-21:
1800 1-day chicks (about 40g) among them were randomly allocated evenly into 10 treatments under a 2 ×5 treatment with 2 anticoccidial programs (Coccidioides and Coccidia vaccine) and 5 dietary Met levels (0.33%, 0.39%, 0.45%, 0.51%, and 0.57%).6 replicates per treatment and 30 chickens per replicate for 21 day.
line 89-93:
1800 1-day chicks (about 40g) among them were randomly allocated evenly into 10 treatments under a 2 ×5 treatment with 2 anticoccidial programs (Coccidioides and Coccidia vaccine) and 5 dietary Met levels (0.33%, 0.39%, 0.45%, 0.51%, and 0.57%).6 replicates per treatment and 30 chickens per replicate (the pen area: 2m*2m=4m2). The trial period was 21 days.
Comment 2:
In addition, cages, welfare, zoohygienic conditions, use of other pharmacological agents in nutrition, etc. should be described. Line 100, 40 samples per treatment - what does it mean? It needs to be developed Line 101, again imprecise, how should this be understood, faeces were collected from the room and separately from the birds from two sections of the digestive tract? Which feces were tested and from how many samples and from what angle? Line 98 onwards, was there approval from the ethics committee to conduct the experiment? Consideration should be given to changing some tables (No. 3, 5 and 7) because they contain too many numbers and thus become illegible. I think that taking into account the above comments will improve the manuscript and make it more readable and understandable. At the same time, I recommend the manuscript for printing.
Answer: We are sorry that we did not describe clearly, and thank you for your valuable comments, which will greatly improve the quality of our article. With regard to fecal collection, we collected fecal samples from the small intestine (Feces are gray, rod-shaped, slightly white-headed) and the cecum (Feces are small amount, red sauce, saccular), respectively, based on the appearance of the feces. Since the experimental design was 30 chickens per replicate, we collected 40 samples per replicate in order to better cover each chicken. The 40 samples were taken from the around and the middle of the pen base on each repeated. Small intestine fecal sample: cecal fecal sample was 4:1(exhibition as fellow picture), and mixed to measure the number of coccidioides oocysts in feces to determine whether the immunization against coccidioides vaccine was successful.
For a better description, we have also added a part of the Material and Methods in line 111-126 as follows:
Feces collection to determine the number of coccidioides oocysts: on the 17th day of the experiment, fresh feces were collected from the around and the middle of the pen base on each repeated (40 samples per treatment in order to better cover each chicken). The fecal samples were collected from the small intestine and the cecum based on the appearance of the feces. Small intestine fecal sample: cecal fecal sample was 4:1, and mixed to measure the number of coccidioides oocysts in feces to deter-mine whether the immunization against coccidioides vaccine was successful.
The ethics committee was added in line 107-108 as follows:
The current study and the use of all broilers were approved by the Laboratory Animal Welfare and Ethics Committee of Xichang University.
The breeding environment was added in line 95-96 as follows:
The temperature of the room was maintained at 30 to 33 °C for the first day and then reduced by 0.5 °C per day. The humidity is maintained at 60-75%.
Our experimental design includes two factors and ten treatments. We want to give as much data as possible through multiple comparisons, principal component analysis, interaction effects, and regression analysis. The table of data display really looks complicated but comprehensive. The figure cannot reflect all the results well, we choose the form of data display. We have added some horizontal lines to the table, which may be easier for the reader to understand.

Reviewer 3 Report
The manuscript has merit for working with a species of bird with little information. The text is not clear in describing the birds and their management. The composition of the experimental diets is shown in Table 1, however, in table 01 is only the experimental design. Table 7, some acronyms are missing.
Author Response
Response to reviewer #3:
The manuscript has merit for working with a species of bird with little information. The text is not clear in describing the birds and their management.
The composition of the experimental diets is shown in Table 1, however, in table 01 is only the experimental design. Table 7, some acronyms are missing.
Answer: Thank you very much for your kind comment. We are very sorry for our careless. We have studied comments carefully and have made correction which we hope meet with approval. We revised the Feed and management in line 80 as follows:
The composition of the experimental diets is shown in Table 2.
We added the acronyms in line 229: CD4+: CD4+T lymphocyte; CD8+: CD8+T lymphocyte;

Reviewer 4 Report
Paper titled „Dietary methionine increased the growth performances and immune function of partridge shank broilers after challenged with coccidia” investigate interesting issue however, some concerns must be elucidated.
Specific remarks
1. lines 60-61 – please give some recent references in this regard
2. line 72 Please notice that your study involved only partridge shank broilers model thus, the results can not be directly extrapolated to other poultry species due to the fact that different species of coccidia appear in different poultry species
3. Lines 90-92 I did not find specific immunization program in the mentioned reference (position 18) – please elucidate
4. Line 116 for what kind of testing samples of ileum were taken
5. Line 126. Nikon light microscope?6. Line 131 – please specify how did you process sample for urea nitrogen measurements?
6. Lines 213-215 This was already discussed in the introduction section, there is no need to repeat
7. Line 224. This reference (position 23) investigated “Vaccines against fungal infections”, Strongly recommend to use some other reports more recent in the chicken model
8. Lines 271-272 vs. 279-280 These sentences seem to be in contrast. Plus, line 287 vs. 305-306. This must be elucidated. According to the present data Met level did not affect most immunological indices investigated herein whereas, weight index of selected immune organs is far too low to conclude that immune functions were improved
9. Line 297. Regarding the novelty; please specify what was of surplus regarding scientific contribution of the present paper in comparison to that mentioned in position 18? (it seems that it is continued experiment), and since most important findings are in contrast, it should be discussed in the present study.
Author Response
Response to reviewer #4:
General Comment:Paper titled „Dietary methionine increased the growth performances and immune function of partridge shank broilers after challenged with coccidia” investigate interesting issue however, some concerns must be elucidated.
Thanks for your kind suggestions and comments, which are all valuable and very helpful for revising and improving our paper, as well as the important guiding significance to our researches. We have studied comments carefully and have made correction which we hope meet with approval. The detail changes are as follows.
Comment 1:
lines 60-61 – please give some recent references in this regard
Answer: Thank you very much for your kind comment. According to your opinion, we have replaced the new references in line 65 (No.14) as follows:
Thomas W. Methionine production—a critical review. Appl MicrobiolBiotechnol.
- 98:9893–9914. DOI 10.1007/s00253-014-6156-y
Comment 2:
line 72 Please notice that your study involved only partridge shank broilers model thus, the results can not be directly extrapolated to other poultry species due to the fact that different species of coccidia appear in different poultry species
Answer: Thank you very much for bringing this point to our attention, which helps to improve the quality of the manuscript. We have modified the statement in line 72-74.
These results might provide partial evidence of dietary Met regulating growth performance and intestinal immune function and its possible mechanisms in partridge shank broilers.
Comment 3:
Lines 90-92 I did not find specific immunization program in the mentioned reference (position 18) – please elucidate
Answer: Thank you very much for your kind comment. We are very sorry for our careless. Your suggestion is very good. The immunization procedure was based on that of yellow feather broiler (Broiler was immunized with Marek's and bursa of Fabricius vaccine on the 1th day. Broiler was immunized with Newcastle disease, infectious bronchitis and H9 avian influenza vaccine on the 7th day, and immunized with H5 avian influenza vaccine by wing muscle injection on the 15th day). We add it to the Material and Methods in line 98-101 as follows:
The broiler was immunized with Marek's and bursa of fabricius vaccine on the 1th day. The broiler was immunized with Newcastle disease, infectious bronchitis and H9 avian in-fluenza vaccine on the 7th day, and immunized with H5 avian influenza vaccine by wing muscle injection on the 15th day.
Comment 4:
Line 116 for what kind of testing samples of ileum were taken
Answer: Thank you very much for your kind comment. We are very sorry for our careless. The sampling position was not clearly expressed. We took 5cm sample of ileal central section. We have also added a part of the Material and Methods in line 127 as follows:
Ileum (about 5cm in ileal central section) was extracted from slaughtered broilers, placed in a dish, cut lengthways, scraped and collected in the same EP tube, and stored in -80℃ liquid nitrogen for testing.
Comment 5:
Line 126. Nikon light microscope?
Answer: Thank you very much for your kind comment. We are very sorry for our careless. We have also added a part of the Material and Methods in line 139 as follows:
The lesion scores range from 0 (no gross lesion) to 4 (most severe gross lesion) used Nikon light microscope (Nikon Eclipse 80i, Nikon Corp., Tokyo, Japan).
Comment 6:
Line 131 – please specify how did you process sample for urea nitrogen measurements?
Answer: Thank you very much for your kind comment. We are very sorry for our careless. We have also added a part of the Material and Methods in line 146-148 as follows:
The concentrations of total protein, albumin and urea nitrogen were measured by ul-traviolet spectrophotometer using the reagent kits (Nanjing Jiancheng Bioengineering, China).
Comment 7:
Lines 213-215 This was already discussed in the introduction section, there is no need to repeat
Answer: Thank you very much for bringing this point to our attention, which helps to improve the quality of the manuscript. According to your suggestion, we deleted this sentence.
Comment 8:
Line 224. This reference (position 23) investigated “Vaccines against fungal infections”, Strongly recommend to use some other reports more recent in the chicken model
Answer: Thank you very much for your kind comment, which helps to improve the quality of the manuscript. According to your suggestion, we looked at a large number of references, but no recent studies, we deleted the original references.
Comment 9:
Lines 271-272 vs. 279-280 These sentences seem to be in contrast. Plus, line 287 vs. 305-306. This must be elucidated. According to the present data Met level did not affect most immunological indices investigated herein whereas, weight index of selected immune organs is far too low to conclude that immune functions were
Improved
Answer: Thank you very much for bringing this point to our attention, which helps to improve the quality of the manuscript. According to your suggestion, we have improved expression and add the result description of multiple comparisons, which may make the article clearer as follows:
In line 287: Met can improve the ratio of CD4+and the value of CD4+/CD8+under two coccidia group (rather than CD8+).
In line 223-225: The results of multiple comparisons can be found appropriate Met can improve the ratio of CD4+and the value of CD4+/CD8+under two coccidia group (rather than CD8+).
The results of Met affecting immune function are not accurate enough. We modified the expression as follows:
In line 306 and 325: We change the “immune function” as “immune function (IgA content in cecal lumina)”
Comment 10:
Line 297. Regarding the novelty; please specify what was of surplus regarding scientific contribution of the present paper in comparison to that mentioned in position 18? (it seems that it is continued experiment), and since most important findings are in contrast, it should be discussed in the present study.
Answer: Thank you very much for bringing this point to our attention, which helps to improve the quality of the manuscript. Our research in 2018 focused more on the study of immune function with Met in two kinds of coccidia treatment, while this study focused more on the study of production. In addition, the two studies focused on different feeding stages. The innovation was to evaluate the Met requirements of partridge shank broilers in two kinds of coccidia treatment, which will be a great significance to future commercial production. At the same time, we also found the effect of Met on the antioxidation level of in two kinds of coccidia treatment, and the study of 21-day feeding stages in partridge shank broilers was scarce. Therefore, this study is also valuable compared with previous studies. In order to improve the discussion part, we add a comparative discussion with the previous research in line 295-299 as follows:
However, in this study, dietary Met did not affect the CD4+ and CD8+, which are same as previous research in our laboratory. The study found that 0.45%Met concentrations treatment has the highest CD4+ cell percentages and CD4+/CD8+ in vaccinated chickens. To our surprise, the results of CD8+ are different which be related to the longer fed cycle may activate the facial system more effectively.
Re-consummated the conclusion
Thanks again for your kind suggestion.

Round 2
Reviewer 1 Report
The authors made a good revision. The version can be accpeted for publication.
Author Response
Thanks for your reviewing.
Reviewer 4 Report
I have no further remarks.
Author Response
Thanks for your reviewing.